# Mapping the Dynamic Complexity of Sexual and Gender Minority Healthcare Disparities: A Systems Thinking Approach

**DOI:** 10.3390/healthcare12040424

**Published:** 2024-02-06

**Authors:** Braveheart Gillani, Dana M. Prince, Meagan Ray-Novak, Gulnar Feerasta, Devinity Jones, Laura J. Mintz, Scott Emory Moore

**Affiliations:** 1Jack, Joseph and Morton Mandel School of Applied Social Sciences, Case Western Reserve University, Cleveland, OH 44106, USA; braveheart.gillani@case.edu (B.G.); meagan.ray@case.edu (M.R.-N.); 2LGBT Center of Greater Cleveland, Cleveland, OH 44102, USA; gfeerasta@lgbtcleveland.org (G.F.); transwellness@lgbtcleveland.org (D.J.); 3School of Medicine, Case Western Reserve University, Cleveland, OH 44106, USA; ljm98@case.edu; 4Frances Payne Bolton School of Nursing, Case Western Reserve University, Cleveland, OH 44106, USA; sem167@case.edu

**Keywords:** SGM, health disparities, healthcare access, intersectionality, complex systems science, minority stress theory, translational science, life course perspective, group model building, healthcare

## Abstract

Sexual and gender minority (SGM) populations experience extensive health disparities compared to their straight and cisgender counterparts. The importance of addressing these disparities is paramount, as SGM groups often encounter significant barriers to accessing comprehensive healthcare, including societal stigma, provider bias, and financial constraints. This study utilizes a community-based system dynamics approach to understand and visualize the barriers to and facilitators of healthcare engagement for SGM groups across their life course. It aims to identify core constructs, relationships, and dynamic feedback mechanisms related to the experiences of connection/disconnection with physical, mental, and dental healthcare of SGM individuals. Barriers to access, such as discriminatory practices and the limited availability of SGM-informed healthcare professionals, exacerbate these disparities, underscoring the urgency of developing targeted interventions. System dynamics, a complex systems science (CSS) methodology, was used for this research. Group model building sessions were conducted with diverse SGM groups, including youth, older adults, and trans and gender-expansive community members. Causal loop diagrams were developed according to an iterative process, and a meta-model of their collective experiences was created. The study revealed extensive, dynamic, and shifting structural barriers for SGM community members accessing healthcare. Societal and structural stigma, provider bias, and pathologization were identified as significant barriers throughout their life course. Community-led interventions and SGM-focused holistic healthcare were identified as critical facilitators of SGM healthcare connection. The findings highlight the need for SGM-affirming and culturally responsive healthcare settings. This paper calls for a concerted effort from SGM health researchers to use CSS in developing interventions to reduce SGM health disparities.

## 1. Introduction

Comprehensive and compelling evidence documents the disproportionate burden of physical and mental health disorders and diseases currently experienced by sexual and gender minority (SGM) groups across the life course [1,2,3,4,5,6,7]. Access to SGM-affirming physical and mental healthcare is one of the drivers of these disparities; however, the complexity of factors impacting access are numerous. Structural- [2,5,8] contextual- [3,7], interpersonal- [3], and individual-level [9] factors are all components of healthcare access. Despite growing research identifying facilitators of and barriers to healthcare access among SGM groups, to our knowledge, the complexity of the problem has not been investigated using a complex systems science approach (CSS). We argue that SGM healthcare access is best conceptualized using CSS for three reasons. First, physical and mental health disparities for SGM groups arise from multiple ecological contexts simultaneously and over time (e.g., school, work, family, peers, and legal structures). Second, bidirectional relationships and feedback mechanisms between the factors and contexts experienced by SGM groups shape their health outcomes. Finally, the stakeholders involved in SGM healthcare access represent disparate fields. For example, the current sociopolitical and legal landscape of state laws being passed to limit or deny access to gender-affirming care involves a myriad of stakeholders, including politicians, special interest groups, SGM advocacy groups, clients, and healthcare and mental healthcare providers [10].

The importance of regional context is another under-investigated aspect of healthcare access for SGM groups [11]. For SGM individuals living in the Midwest and the South, regions with greater restrictions on affirming care and fewer protections for sexual and gender minorities, the issue of healthcare access is even more pronounced. The Midwest United States and especially Ohio are focal points for legislative action affecting gender-affirming medical care for transgender youth. Ohio House Bill 68, which effectively bans all gender-affirming care for minors under the age of 18, recently passed the Ohio State Senate. Ohio joins 21 other states, the majority in the Midwest and South, that have passed legislation to ban gender-affirming care up to age 18 [12,13]. These harmful legislative actions are a stark example of the regulatory and legal challenges facing SGM healthcare access.

This translational health pilot study involves a purposive sampling of 27 individuals encompassing a wide spectrum of demographic diversity, including youth, elderly SGM individuals, and those identifying as transgender or gender-expansive (TGD) from a Midwestern city’s LGBT center. The primary objective of this study is to explore and articulate the intricate dynamics governing healthcare connection and disengagement within these communities, ultimately striving to forge a more inclusive, empathetic, and effective healthcare system. By integrating advanced systems science methodologies, this research contributes significantly to the broader discourse on decreasing healthcare disparities within the SGM community, a crucial step toward global healthcare equity. 

In the present study, we perform an exploratory analysis of the complexities surrounding the healthcare disparities faced by SGM groups. Utilizing a community-based system dynamics approach, we delve into the multifaceted nature of these disparities, mapping out both the barriers and facilitators influencing different SGM groups’ healthcare access. Therefore, this study aims to contribute to the development of more effective, inclusive, and equitable healthcare strategies for SGM communities. This study positions itself at the forefront of intersectional and life course theories while using CSS, endeavoring to enhance our understanding of healthcare engagement within SGM groups across various life stages.

### Intersectionality, Minority Stress, and Disconnection from Care across the Life Course

Our study is grounded in the theory of intersectionality and the minority stress framework. Intersectional approaches to health disparities research are gaining in popularity [14]. Intersectionality theory centers the experiences of structurally and socially marginalized groups, e.g., “decentering” whiteness and maleness [10,15,16], and promotes mapping as an analytic tool to name and trace the institutional processes that synergistically work to oppress marginalized populations [17,18,19]. Intersectionality and related bias are well documented in SGM health literature [20]. Disclosure of one’s sexual orientation or gender identity is often critical for appropriate healthcare provision; however, studies indicate a consistent lack of transparency from patients in critical healthcare units such as oncology [19,21], palliative care [22], and mental health [23]. 

Researchers in this area currently call for multilevel modeling to demonstrate the impact of structural inequality on individual experiences of multiple forms of marginalization [24]. CSS meets this demand to analyze identity in healthcare connection and disconnection for SGM individuals, filling an important gap in the extant research.

Intersectionality is also increasingly applied to the minority stress model (MS) to capture how SGM individuals who experience multiple forms of marginalization based on social group membership (e.g., race, ethnicity, gender identity, and socioeconomic class) are differentially impacted by minority stress. MS shows how SGM-specific stressors at various levels (e.g., institutional, interpersonal, and intrapersonal) combine to inform health and mental health outcomes [22]. For example, heteronormative and cisnormative health policies that exclude or silence SGM access to affirming healthcare and mental healthcare, compounded by experienced discrimination, negatively influence mental health [9]. 

Finally, concepts from Elder’s life course theory, including cohorts (age cohort/generational cohort), linked lives (how interpersonal relationships affect developmental trajectories), transitions and turning points, and human agency, are useful in understanding how SGM groups navigate healthcare access. Transitions and turning points are especially salient to SGM lives because they go beyond normative ideas of “life stages” and “milestones” based on heteronormative constructs [25]. Collectively, minority stress, intersectionality, and queer life course theories guide our study of SGM connection and disconnection from healthcare across the life course. By visualizing the mental models of SGM groups at different life course stages, this study identifies how various components of the healthcare structure remain the same while others change as an SGM individual moves across their life span.

## 2. Method

Complex systems science (CSS) examines how collective patterns (such as population-level racial and ethnic health inequities) [8] are derived from ever-evolving interrelationships among individual parts of a system (such as how people dynamically interact with each other and their environments) [8]. 

System dynamics is a CSS methodology that utilizes causal thinking and a focus on how problems change over time to model, analyze, and improve social systems [10]. It allows us to analyze the dynamic and complex social forces that shape health disparities among minoritized groups, including long-term causal relationships, conflicting goals, and stakeholder interests [10]. In this study, we apply community-based system dynamics (CBSD), a CSS approach grounded in participatory research methods, to understand how SGM groups engage with healthcare, including mental, physical, and dental care, across the life course. The specific aims were to identify the dynamic patterns of factors related to seeking out, connecting to, or avoiding and disconnecting from healthcare. Group model building (GMB) is a structured and collaborative problem-solving approach involving bringing together community members, experts, and academics to address complex and multifaceted problems collectively [26]. It allows participants to contribute their knowledge and insights, which may not be available from other data sources [24]. GMB was used with four community groups (older adults, young adults, transgender and gender-diverse (TGD) individuals, and center staff) from a mid-sized Midwestern city’s LGBT center. A detailed process map of the GMB sessions, including the scripts and exercises used, is provided in Figure 1. The developed causal loop diagrams (CLDs) about connection and disconnection to healthcare were elicited from SGM community members using a series of convergent and divergent exercises [27]. All the scripts were acquired from Scriptapedia, a global, open-source platform for group model building exercises [28], and modified to comply with the restrictions imposed due to COVID-19. 

### 2.1. GMB Community Participation

Community member input was prioritized in the CLD development. Care was taken to retain the language and voice used by the participants within each CLD [29]. Multiple iterations and re-design of the CLDs during and between several GMB sessions resulted in the identification and synthesis of the underlying structures impacting the SGM community’s connection with the healthcare system. Separate GMB sessions across identity groups (age, gender identity, provider) allowed each group to have their own voice in visualizing and articulating their specific mental models related to healthcare access. CLDs of different groups were later shared with participants across groups, thus enhancing their sense of connection, and the resulting meta-model served as a unifying boundary object—a visual artifact for all the stakeholders to use as a means of communication, collaboration, and agreement across time [30] for the total sample [31]. To our knowledge, this is the first application of SD techniques to the healthcare gaps and connections experienced by SGM populations.

Participants. We recruited 27 individuals between 11 February 2021 and 29 October 2022 to participate in the GMB sessions. Participants were recruited through a formal research partnership with an LGBT community center in a small Midwestern city. Participants 18 or older who self-identified as a sexual and/or gender minority were eligible to participate.

Recruitment. Purposive sampling was used to identify four groups from which to recruit. SGM older adults (65+), transgender individuals, young adults (18–24), and LGBT center staff comprised these groups. The members of the study team conducted initial Zoom visits to introduce the project and invite individuals to participate. Interested individuals were emailed the informed consent document before the GMB session. The GMB sessions were then scheduled with all four identity groups. The participants provided verbal informed consent at the beginning of the initial sessions. The participants received a $50 gift card for the first GMB session and a $25 gift card for participation in the second. 

### 2.2. GMB Sessions

Three facilitators conducted each GMB session. Session 1 focused on community building, variable elicitation, and developing an initial CLD based on the factors causing disconnection to healthcare. The CLD was refined by the core modeling team between sessions 1 and 2 using a review of the recorded sessions and notes taken during sessions. GMB session 2 included reviewing and providing feedback on the modified CLD, focusing on the factors connecting the participants to healthcare. The participants iterated over the model until they expressed satisfaction, believing it had reached saturation and aligned with their mental models. A meta-model was produced by combining the 4 CLDs developed in the group sessions. A select team of participants reviewed the final model sets for session 3, where the meta-model was presented for iteration and validation. All sessions were conducted using Zoom, which aligned with the COVID-19 protocols in place during the study period. The Case Western Reserve University Institutional Review Board approved the study (STUDY20200359).

### 2.3. Analysis

Throughout this project, participants were explicitly urged to provide feedback to the modeler (author 1) and modeling facilitators (authors 2 and 3) in real time. Input from the participants was actively validated and affirmed. This encouraged active engagement from the participants in the modeling process to provide rich and nuanced narratives for the CLD, allowing for a collaborative and iterative approach to refining it. 

The core modeling team further distilled the groups’ CLD after each session. This process involved authors 1, 2, and 3 individually reviewing the Zoom video recording and validating constructs and relationships. The core team then met to discuss these reflections and resolve any discrepancies between perspectives. These revised CLDs were shared with the community group in a second meeting for validation, clarification, and any additions. After completing both GMB sessions, the core research team engaged in a second analytic process to distill the individual CLDs from each SGM group. To develop the meta-model, the team individually reviewed all of the community-validated CLDs to identify core elements across the groups. According to multiple analytic iterations, each of the constructs of the individual CLDs was tabularized, and the core themes were articulated. The modeling team then discussed each construct to identify similar themes across the CLDs. Across the CLDs, common barriers and facilitators related to healthcare connection and disconnection were identified. These common constructs and relationships were extracted and developed into the meta-model CLD. The modeling team went through several rounds of consensus coding to generate this model. A final member check focus group was performed for the meta-model by sharing it with representatives of each GMB session for validation.

## 3. Results

The participant demographics are provided in Table 1. Aggregate participant details are provided to protect participant anonymity. Cisgender women were the highest represented in this sample, followed by transgender women, cisgender men, genderqueer individuals, and transgender men. Most of the sample identified under the umbrella term queer (gay, lesbian, pansexual, or queer). Slightly over half the sample identified as white. There was a wide age range (19–82) and a high standard deviation of 19.8. 

Figure 2, Figure 3, Figure 4 and Figure 5 show the final models derived from the GMB sessions with the center staff, older adults, young adults, and the transgender and gender-diverse group, respectively, with the essential causal loops identified and explained in Table 2. To read these causal models, consider that the lines connect two factors only. Each arrow connector represents one causal relationship. Causality in this method refers to how participants understand and experience factors as linked, including how one factor informs another. The solid-lined arrows indicate an explicit relationship (stated directly by participants), and the dotted-lined arrows indicate an implicitly stated but community-confirmed relationship. A negative sign at the arrowhead articulates an inverse relationship. Causal loops are expressed with a circular arrow, where an R within the loop defines a reinforcing loop while a B within a loop defines a balancing loop. A reinforcing loop is a closed loop amongst two or more variables in which a change in a variable leads to further changes in the same direction. When a variable increases (or decreases) within the system, it triggers a process that amplifies the exact variable change, leading to exponential growth or decline within the system. Reinforcing loops often result in the escalation of a particular phenomenon, creating a self-reinforcing cycle. This can be either beneficial, leading to growth and improvement, or detrimental, causing instability or decay within the system. In contrast, a balancing loop is one in which a change in a variable triggers processes that act to counteract or dampen that variable change, thereby maintaining stability or equilibrium within the system. Balancing loops work to resist and offset deviations from the dynamic equilibrium, ensuring that the system returns to a stable condition after disturbances. They are essential for maintaining a system’s overall stability and preventing extreme fluctuations.

Figure 6 shows the final integrated meta-model, highlighting the combined barriers and facilitators for all the groups. The constructs in the meta-model are detailed in Figure 7. The meta-model factors of emotional and psychological violence, provider bias, pathologization, intersectional oppression, and marginalization were the primary reasons for disconnection from healthcare across the life course. The meta-model also shows great strength and robustness within the SGM community. The model suggests that community resilience is developed via imaginative coping mechanisms, reinforcing the need for community-generated interventions. Many examples of such interventions decrease a community member’s disconnection from healthcare and simultaneously support structural changes toward the development of SGM holistic healthcare [28,30,32]. Many community members preferred engagement with such holistic healthcare and identified it as a facilitator for connection to healthcare. The meta-model identifies four significant clusters of factors present across all the groups. These include stigmatization, mental well-being, provider violence and disconnection from healthcare/holistic healthcare, and community-generated interventions. We discuss each of these significant cluster areas in turn. 

### 3.1. Stigmatization

Dimensions of stigmatization, identified in the meta-model according to the constructs of intersectional oppression, pathologization, marginalization, and emotional and psychological violence, were present across all groups. All the groups discussed oppression due to their SGM identities, with the youth and older adults additionally discussing marginalization based on age. Pathologization was a shared experience across the life course of all the focus groups, with the older adults having experienced pathologization due to their SGM identities in their youth in different ways than the pathologization experienced by current youth. Pathologization was manifested across the groups, including denial of their identities and experiences and fearing conversion therapy for the youth group; legal and health discrimination for the transgender group; legal discrimination for the center staff; and medical discrimination for the older adults. All groups reported experiencing various aspects of emotional and psychological violence, with the center staff highlighting the voyeurism and fetishization of SGM bodies and older adults, noting the negative impacts of harmful messaging from various social and political contexts.

### 3.2. Mental Well-Being/Emotional Load/Resilience

Participants reflected on how different manifestations of queerphobia often thwarted their personal journeys toward mental well-being. The older adults emphasized the power of self-advocacy and community advocacy as pathways to their mental and emotional well-being. This was mirrored by the youth group, who highlighted peer-to-peer support as a strategy for building resilience in the face of stigmatization and improving their well-being. The center staff and TGD group underscored the emotional load experienced by SGM individuals over their life course, whether due to direct stigmatization or the social and financial oppressive structures that are byproducts of societal queerphobia. The TGD group identified additional legal and criminalization factors that increased the emotional burdens they experienced and forced them to navigate complex systems to access essential services. The construct of resilience also emerged in each group, with the importance of hope and knowledge transfer across generations emerging as a critical source of strength for the entire community.

### 3.3. Provider Violence/Disconnection from Healthcare/Holistic Healthcare

Provider violence emerged as a common construct within the modeling sessions, with members of each group sharing stories of the mistreatment and harm perpetuated by healthcare providers and systems. This continuous mistreatment, whether via microaggressions such as misgendering or blatant acts of discrimination such as the denial of care based on gender identity or sexual orientation, elicited examples of various coping mechanisms across different groups. The youth group sought to find providers with as many overlapping intersectional identities as possible to minimize trauma and increase their comfort in accessing healthcare services. The center staff members reported instances of misconceptions within healthcare providers about homosexuality being caused by trauma and of the objectification and fat-shaming of transgender individuals. The staff underscored the need for holistic healthcare approaches that address the unique needs of SGM individuals, including mental health support, gender-affirming care, and cultural competence. Members of the transgender group discussed pivoting to “black market” sources in seeking gender-affirming care due to their distrust of and subsequent disconnection from the standard healthcare model. Additionally, many older adults mentioned hiding essential aspects of their identity and “staying in the closet” to continue receiving healthcare. All groups emphasized the value and need for holistic healthcare options as an alternative to the standard healthcare model.

### 3.4. Community-Generated Intervention

All the modeling groups identified the high value of formal and informal community-generated interventions as a buffer to the stigmatization experienced and to facilitate their well-being. These interventions included formal support groups; peer-to-peer mentorship; the referral of safe healthcare providers and additional resources between members; healthcare information-sharing, including safe and newer strategies for hormone therapy; sharing clothing, shoes, and other gender-affirming items; and the development and nurturing of safe and affirming spaces within the community for individuals to gather and connect. Community-generated interventions were particularly prominent in the TGD group, where participants identified the value of specific transgender older adults who were “family” for the newer members of the group and served as mentors and guides in navigating various challenges and accessing resources.

## 4. Discussion

This translational health pilot study applied CSS methods to generate emic knowledge of the social, environmental, and structural mechanisms within healthcare connections among SGM community members residing in a mid-sized Midwestern city. We engaged diverse SGM individuals across the life course in GMB sessions to describe how connection and disconnection to physical and mental healthcare work from an endogenous, feedback-based perspective over an SGM individual’s lifetime [8,14,15]. The study findings highlight the emergent intervention strategies developed by SGM community members to manage their personal and community health, well-being, and relationships with healthcare. Our study findings are consistent with the growing literature on SGM healthcare navigation [33]. However, our research provides a CSS perspective that animates how various factors simultaneously affect connection and disconnection from care across the life course (Figure 2, Figure 3, Figure 4, Figure 5 and Figure 6). The feedback loops identified within various CLDs show that multiple barriers prevent SGM community members from successfully navigating the healthcare system and contribute to a cycle of disconnection and mistrust, leading to further disengagement and disconnection from healthcare services. The key takeaways from the research include intersectional barriers to care across the life course, provider engagement, and community-based interventions.

### 4.1. Intersectional Barriers to Care across the Life Course

A growing body of literature confirms the impact of intersectional, specifically SGM-identity-related, stigma on the physical and mental health outcomes of SGM individuals [34,35,36]. Repeated studies have confirmed the impact of the stigmatization of SGM individuals, particularly SGM youth, across their life course on their health and mental health [22,35]. SGM youth are more likely to experience mental health issues, substance abuse, and suicide than their heterosexual counterparts [37]. Older SGM individuals face compounded disparities due to ageism and isolation, highlighting the importance of the life course perspective in understanding their health disparities [35,36,37].

The SGM groups in our study confirmed these findings by articulating the specific intersectional (age and SGM status) barriers they experienced while accessing healthcare. They shared how providers often dismissed their invisible (sexual) identities or avoided asking them about their sexual orientation altogether, leading to missed opportunities for sexual healthcare. Additional stigmatization of SGM youth in our study included efforts on behalf of providers to “cure” their orientations by suggesting methods such as reparative therapy to their parents (Figure 5). These findings have been confirmed by extant research, which indicates several barriers to care experienced by SGM groups, including gatekeeping, the pathologization of SGM identities and behavior, and the policing of queer bodies, especially at the intersections of race, gender, and class [9,29,38]. Other similar aspects of stigmatization, such as dismissal or erasure of sexual orientation and gender identity, were articulated by the SGM youth participants and the older adults, indicating that such structures of stigmatization remained constant across their life course (Figure 5). Many older adults articulated specific forms of stigmatization they had experienced based on historical time and place, such as their same-sex relationships being considered illegal and the explicit pathologization of their sexual orientation by the Diagnostic and Statistical Manual of Mental Disorders (Figure 3).

### 4.2. Provider Engagement

The importance of provider engagement with SGM patients to facilitate their connection to the appropriate healthcare cannot be understated [39,40]. SGM-affirming care, inclusive of clinical settings (e.g., display of SGM symbols), forms (e.g., inclusive language for gender identity, chosen names and pronouns), staff, and providers, is critical to ensure patient/client connection, safety, and thus engagement in treatment [29,38,41,42,43]. In our research, the participants explicitly identified community-trusted providers and discouraged newer community members from engaging with providers who were known to be harmful. The TGD group described trustworthy care providers as those who took the time to listen to their patients, neutralized the power dynamic by sitting next to patients, were available to patients in moments of crisis, advocated for their patients, and stood up to transphobia perpetuated by their colleagues. Additionally, the participants stated that they avoided mainstream healthcare providers when acquiring hormone replacement therapy and gender-affirming surgery due to past traumatic experiences with providers and healthcare settings. This finding is in line with other studies that show how TGD individuals avoid healthcare due to fears and actual instances of structural and individual discrimination and silencing [44,45].

The SGM youth participants had greater expectations of providers. They sought concordant identities with their providers and indicated their awareness of the financial burdens of healthcare and their multiple marginalized identities as dissuading factors when engaging with healthcare. Racially minoritized youth participants shared narratives of juggling provider preferences and disclosing their identities in ways their white racial/ethnic counterparts did not (Figure 5). The importance of identity concordance is substantiated in existing research which identifies the importance of diversity within healthcare providers to match patient populations [46,47]. This has been verified within the realm of mental health, where a provider’s cultural background and values are critical to the development of the client–therapist relationship [48].

Older adults shared narratives of rejection and disrespect from their providers, often pointing to decades of damaging laws and policies, negative media campaigns, and harmful policies as causing many of the hardships they had to navigate (Figure 3). This finding is confirmed by aging literature, indicating that older adults experience a lack of respect and dismissal of their voices within Western healthcare systems [49,50]. Such barriers can be even worse for people who are additionally marginalized because of gender identity, sexism, racism, or ableism.

The similarities between the older adults and the youth were compelling. Both groups spoke of ageism (either due to being too young or too old), leading to a lack of respect and autonomy within the medical model. Additionally, they both spoke about the deleterious impacts of provider bias and the pathologization of their identity. These similarities and differences between the older adults and the youth describe SGM health disparities as a dynamic problem; over time, the shape of the problem and the community’s needs are changing. 

### 4.3. Community-Based Interventions

Post-traumatic resilience within the SGM community [51], with aspects of both individual and community resilience, is increasingly recognized as an essential factor in understanding the experiences of SGM individuals [52]. Community-based interventions such as peer mentoring and across-generation mentoring groups, identity-based support groups, culturally competent mental health services for adults, and age-appropriate support groups and activities for older adults have been identified as potential strategies to address these disparities [39,40,51,53]. In our study, participants shared the criticality of community-based interventions in leading individuals to connect with healthcare. For example, within the older adult session, participants shared how community-based recovery groups (such as Alcoholics Anonymous) had been critical to their survival, a fact that has been supported by existing research [54,55]. Community members also identified support groups as a space to share advocacy tips and skills, knowledge about providers and, finances, and other topics related to well-being. Older members citted the existence of an underground “lesbian health network”, which connected community members with feminist practitioners who were aware of sexism, racism, and homophobia in the lives of Black lesbian women. Support groups, whether focused on addiction or providing a safe space for individuals to receive peer support, have been identified as substantial factors in SGM health and welfare by existing research [39,40]. 

The members of the transgender and gender-diverse sessions articulated the presence of families of choice, where members who were more settled within the community shared knowledge (job opportunities, safe doctors, sources for hormones) and resources (including but not limited to jewelry, clothing, and housing) with the newer members of the community, hence sharing data across cohorts and generations (Figure 4). Transgender and gender-diverse individuals using “chosen family” as a space of refuge and support is supported by the current qualitative literature [53] and is now confirmed within groups. Community networks were prominent within the GMB groups in this study. Participants shared the names of helpful and harmful care providers, emerging technologies for hormone replacement therapy, and strategies for navigating the bureaucracy of insurance policies. These findings are in line with a growing body of research that shows the importance of community connection and social support for SGM individuals across the life span as related to both physical and mental health and navigating barriers presented by ostracization and discrimination due to their identities [51,52,56]. However, our CSS analysis indicates that many of these interventions are required concurrently across the life cycle to effectively reduce and mitigate the negative impacts of stigma and discrimination on the physical and mental health outcomes of SGM individuals.

While this study offers significant insights into the healthcare disparities experienced by SGM individuals, it is essential to acknowledge its limitations. Firstly, the use of a community-based system dynamics approach, while robust in capturing complex interactions, may not account for all the nuances of individual experiences, particularly those at the intersection of multiple marginalized identities. This study is focused on the group-level dynamics experienced by identities rather than by specific individuals. Additionally, the study’s focus on a specific Midwestern community may limit the generalizability of its findings to other regions or demographic groups. 

Despite these limitations, the implications of this research are far-reaching. It highlights the urgent need for healthcare systems to adopt more inclusive and SGM-affirming practices, particularly in mental, primary, and specialist healthcare. The study underscores the necessity of holistic healthcare models that are responsive to the unique needs of SGM populations, including addressing societal and structural stigma, provider bias, and the pathologization of SGM identities. The insights gained from this research can inform the development of targeted interventions and policies aimed at reducing health disparities and improving the overall well-being of SGM individuals.

Moreover, this study contributes to the broader discourse on healthcare equity, emphasizing the role of systemic and community-level factors in shaping healthcare access and the quality of care for marginalized groups. It calls for a concerted effort from healthcare providers, policymakers, and researchers to work collaboratively toward dismantling barriers and creating more equitable healthcare environments for all, particularly for those within the SGM community. It is acknowledged that when healthcare access is freely accessible to the most marginalized members of our communities, the level of care for everyone benefits. It is recommended that future studies focused on healthcare access disparities in SGM communities use additional CSS methodologies and system dynamics simulation modeling to identify context-specific leverage points to alleviate this crisis. 

## 5. Conclusions

This study identifies many critical factors experienced by SGM individuals within various stages of their life course that limit consistent engagement with healthcare and promote SGM health disparities. It also identifies a series of community-generated interventions that help SGM individuals connect to healthcare. A critical finding of this study is that no singular intervention strategy within any salient context (e.g., family attachment therapy; school-level anti-bullying campaigns [57], or identity-based social support groups [58]) will “fix” the problem of structural and systemic SGM-bias, stigma, discrimination, and oppression [59,60,61]. SGM healthcare barriers work together as reinforcing obstacles across the life course of individuals. Therefore, community-based supports and connections must also work in concert with each other, adapting to an individual’s needs as they travel through their life course and gain and lose privileges (such as in health, age, community support, and financial support) in the process. This study narrates the distinct experiences and obstacles experienced by different intersectional groups within the SGM community. For example, the final set of CLDs indicated that the TGE CLD had the most harmful reinforcing loops and critical variables, pointing toward a crisis for the TGE community and a high amount of transphobia. Most TGE group members identified with multiple marginalized identities, further promoting the harmful loops of exclusion they experienced from the healthcare system.

This study underlines the value of using community-based system dynamics with marginalized populations to impart system thinking skills and techniques to them. Minoritized individuals often use systems thinking to navigate bureaucratic and ineffective systems, which are ill designed to support their needs. In our study, community members, especially those who organized the GMB sessions, became well versed in reading the CLD conventions, sharing them with their peers and articulating other systems in their lives using CLDs. 

Ultimately, there has been a rise in SGM health interventions in recent years. However, for these interventions to be effective, they must work together cohesively to rapidly address the health disparities experienced by SGM individuals. To achieve this, CLDs like the ones in this study should be created to map out multiple interventions, their impacts, and their unintended consequences. To move toward equity, the medical model must adopt a systems thinking approach that prioritizes marginalized populations, including SGM individuals. This means centering the experiences and knowledge of those whom the system has historically underserved.

## Figures and Tables

**Figure 1 healthcare-12-00424-f001:**
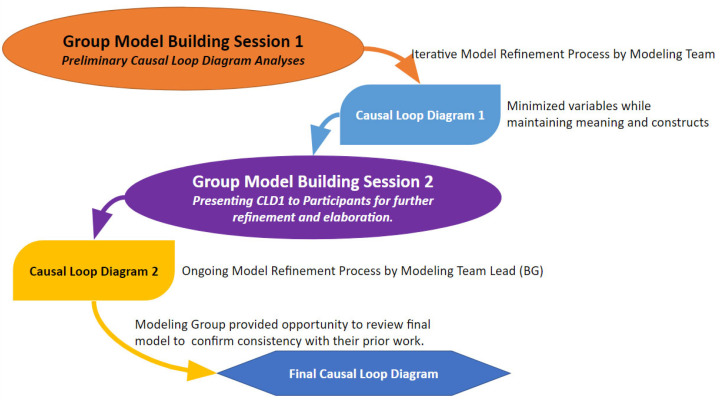
Group model building process for model creation.

**Figure 2 healthcare-12-00424-f002:**
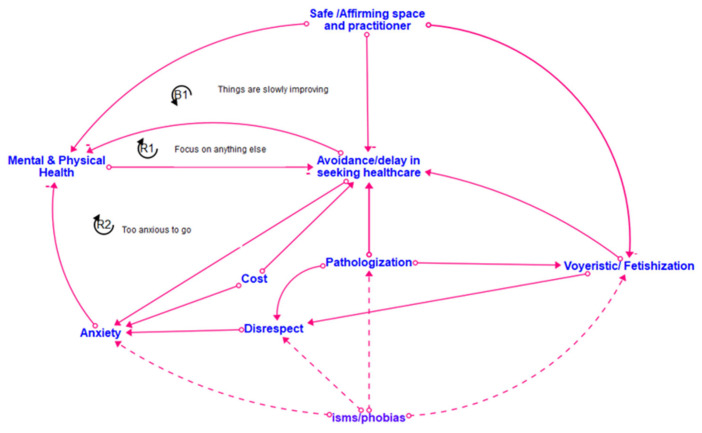
Center staff CLD.

**Figure 3 healthcare-12-00424-f003:**
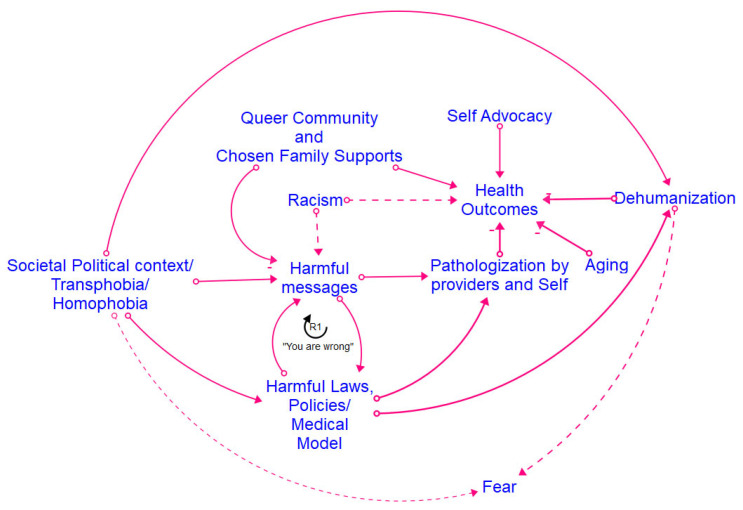
Older adult CLD.

**Figure 4 healthcare-12-00424-f004:**
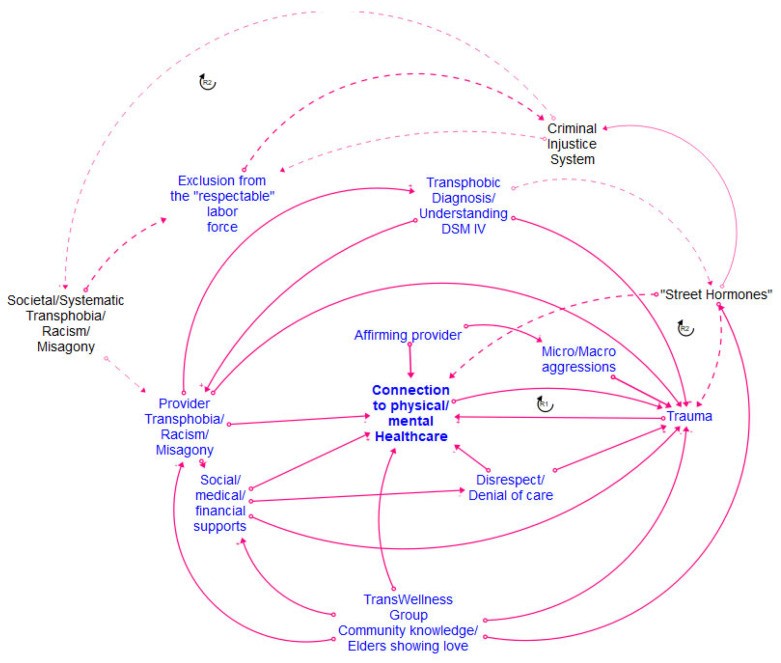
Transgender adult CLD.

**Figure 5 healthcare-12-00424-f005:**
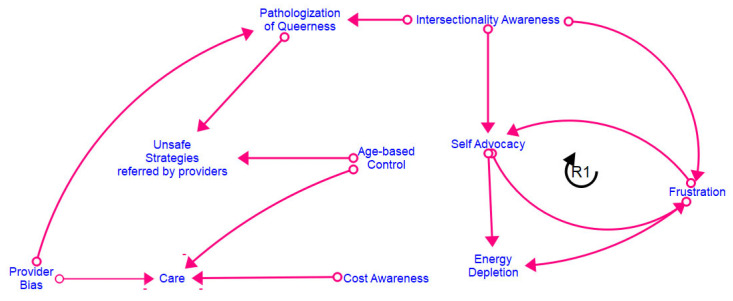
Youth CLD.

**Figure 6 healthcare-12-00424-f006:**
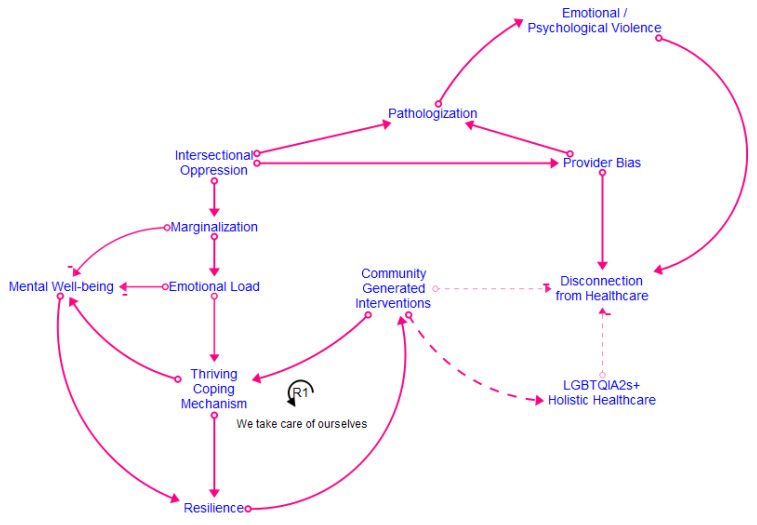
Meta-model.

**Figure 7 healthcare-12-00424-f007:**
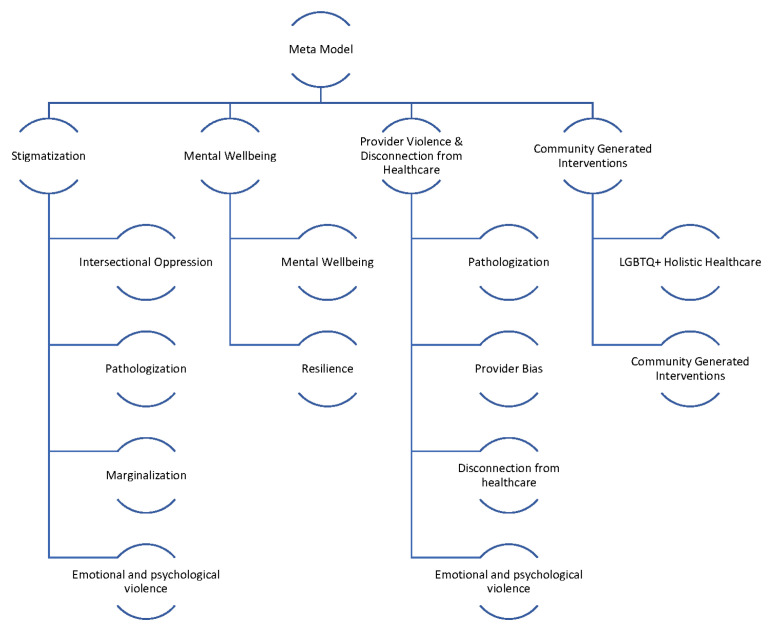
Meta-model constructs.

**Table 1 healthcare-12-00424-t001:** Demographic characteristics of participants at (*N* = 28).

Demographics	*N*	%
Gender identity		
Cis woman	8	28.5
Cis man	6	21.4
Trans woman	6	21.4
GNC/NB/GQ	4	14.2
Trans man	4	14.2
Sexuality		
Lesbian/Gay	11	39.2
Heterosexual	6	21.4
Queer	3	10.7
Pansexual	5	17.8
Bisexual	2	7.15
Questioning	1	3.5
Race/Ethnicity		
White	16	57.14
SWANA	7	25
Black	3	10.71
Biracial	2	7.1
Age		
18–25	7	25
26–40	7	25
41–60	6	21.4
60+	8	28.57

**Table 2 healthcare-12-00424-t002:** Empirical Pathways and Feedback Loops in Meta CLD with Verification across all CLDs.

Definition	Constructs from Individual Group CLDs That Inform the Meta Model Constructs
**Intersectional Oppression** **Pathway:** *Intersectional Oppression* → Provider bias b. Pathologization. C. Marginalization	
The overlapping and interrelated systems of oppression experienced by SGM individuals who belong to other marginalized groups (e.g. BIPOC, disabled, immigrant, non-Christian religion, body size, age	***Older adults CLD*: Societal Political Context/Trans- and homo-phobia; Racism** *“The prevailing attitude when we were all coming of age was that being LGBT was wrong, unhealthy, it was illegal.”*
**Provider Bias** *Pathway: Provider Bias* → Pathologization	
Conscious or unconscious attitudes, beliefs, or stereotypes that impact clinical assessment and/or treatment related to being a sexual and/or gender minority	***TransWellness CLD*: Provider transphobia, racism and misogyny; R4 “*Trained to Hate Me*”** *“The doctor wouldn’t see me because I’m trans. And the doctor wouldn’t even prescribe me. He wouldn’t prescribe me an EpiPen…He wouldn’t even come and see me. Yet he charged me a $500 bill.”*
**Pathologization** **Pathway:** *Pathologization* → Emotional/physical Violence	
Moral/religious or medical pathologizing of SGM bodies and identities	***Youth CLD:* Pathologization of Queerness** *"...[it] made that internalized homophobia that much more difficult to work through. It just sort of made me feel like I wasn’t deserving of health care because I was a queer."*
**Emotional-Physical Violence** **Pathway:** *Emotional/physical Violence* → Healthcare Disconnection	
Physical, emotional, or psychological aggressive attacks on personhood	***Older adults CLD*: Harmful messages about being SGM** *“The prevailing attitude when we were all coming of age was that being LGBT was wrong, unhealthy, it was illegal…perverted.”*
**Disconnection from Healthcare** Explanation: The final product of Intersectional oppression, Provider Bias, Pathologization, and Emotional-Physical Violence	
The outcome of historical, systemic, and lived experiences of exclusion, discrimination, bias, and pathologization of SGM individuals and communities that leads to distrust, avoidance, fear, and anticipated maltreatment in healthcare settings.	***TransWellness CLD*: R2 “*Too anxious to go*”** *"…I haven’t been to a doctor’s office. Do not feel safe in the doctor’s office for any medical issues that I’ve had recently or needed help with."*
**Marginalization** **Paths:** *Marginalization* → a. Decreased wellbeing b. Increased Emotional Load	
SGM individuals and communities are denied access to resources, power, and status in mainstream social, economic, and political systems vis-à-vis discrimination	***Youth CLD:* Age-based control of healthcare access and decisions** *“…it’s a running thought in my mind and a running conversation that I continually have with myself that if I felt safe in a therapist’s office, could I be avoiding taking a medication altogether? It’s like a feeling based. If I felt safe … A huge chunk of my anxiety revolves around my gender and sexuality like I’m sure it is for a lot of people. Could that be avoided?”*
**Emotional Load** *Emotional Load*→ a. Decreased Mental wellbeing b. Decreased Thriving Coping	
The mental, psychological, and emotional burden experienced by SGM individuals due to societal stigma, expectations, and cultural norms. This load can be highly taxing on the individual’s mental and physical health leading to burnout, exhaustion, and a reduced ability to cope with daily stressors.	***Staff CLD*: Anxiety** *“I hadn’t had a dentist appointment in four years, I want to say. And I knew those whole four years that I had those cavities, because I knew what the cost was going to be. And it was at the top of my, my to do list. And knowing that it was at the top of my to do list was given me a lot of anxiety. Like I need to get this done. I need to get this done. My teeth are going to fall out. I can’t, I can’t get a boyfriend if my teeth fall out, you know. “*
**Thriving Coping Mechanisms** *Paths: Thriving Coping Mechanisms* → Resilience	
Wellbeing supportive strategies used to manage SGM-based stressors (e.g., discrimination, pathology, bias, rejection) across intra-, inter-, and systems-level sources.	***Older adults CLD*: Self Advocacy** *“I’ve only had minor problems with discontinuity because I know I need the health care. And I seek it. And if it’s inappropriate, I keep seeking till I find, for myself, what is appropriate in regards of a care provider.”*
**Mental Well-Being** *Mental Wellbeing* → Resilience	
Overall psychological, emotional, and relational health and satisfaction, including developing thriving coping mechanisms.	***TransWellness CLD*:** social, medical and financial supports “*Okay, so here’s what I want you to do. This worked for me. Talk to your counselor. Say you need a medical marijuana for your PTSD because it’s gotten out of control. Because that worked for me, and they will direct you to a place. I can actually send you a link to [name] network. That’s who did my intake as well I got in fairly quick. And everything went really smoothly. So once we’re done with this chat, I’ll send you that info. Okay*?”
**Resilience** Resilience → Community Generated Interventions	
An individual’s ability to cope with and adapt to adverse situations, stress, and challenges in a healthy, positive way. It involves the capacity to recover from difficulties and bounce back from setbacks, stress, and trauma.	***Youth CLD:* R1 “*I won’t give up*”** *“Well can I get my discharge papers in? I’m trying to leave. Like y’alll night racking up a bill, and y’all not even going to help me. I didn’t even get this medicine that y’all said y’all were going to give me. So, let me just bounce. (and seek help elsewhere) “*
**Community Generated Interventions** Paths: *Community Generated Interventions* → a. Increased Connection to Healthcare b. LGBTQIA2S + Holistic Healthcare	
Community members identify areas of need and develop/implement informal and formal strategies (e.g., mutual aid; anti-bias awareness programs; activism) to overcome systemic barriers, build affective bonds and relationships, and disrupt unequal systems.	***Staff CLD*: Safe/Affirming space and practitioner** *“And then they are, you know, talking to me, and you know, I’m feeling good, because I’m in a conversation with the doctor that I like.”*
**LGBTQIA2S+ Holistic Healthcare** Paths: LGBTQIA2S+ Holistic Healthcare → a. Increased Connection to Healthcare	
Holistic healthcare is an antidote to the dominant White, western-culture, andro-centric, medical model. It recognizes the interconnections of physical, emotional, mental, social, cultural, and spiritual health. LGBTQIA2S+ individuals as whole and not abnormal/unhealthy.	***Older adults CLD*: Harmful laws/policies/medical model** *“For every one of those points that we have stigmatize or pathologize LGBTQ people, there is a back door that brings us right back to community belonging and a LGBT community Center of some way shape or form, formal or informal. And it all gets us right back to what you put in. Reidentification of who we are which in turn builds our self-esteem. And makes us, allows us, not makes, allows us to become integral parts of successful living and successful society.”*
**Feedback Loop R1**: *“We take care of ourselves.”* Community Generated Interventions →+Thriving Coping → +Resilience → + Community Generated Interventions	
A reinforcing positive loop where increased community generated interventions, thriving coping, and resilience are supportive of and strengthening to one another.	** *Transwellness CLD* ** *“And so, for the exception of the newer girls, most of the girls in the community know of me. But a lot of them know me personally. And I’ve, you know, tried to help them in transitioning and rough. It’s, it’s rough. Even today, it’s still rough.”*

## Data Availability

The data supporting this study’s findings are available from the corresponding author upon reasonable request.

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
