# Peer review of "Mapping the Dynamic Complexity of Sexual and Gender Minority Healthcare Disparities: A Systems Thinking Approach"

_healthcare, 2024, doi:10.3390/healthcare12040424_

Round 1

Reviewer 1 Report

Comments and Suggestions for Authors

This study reports the experiences of marginalized groups in the context of health care in the US. The article relies on model-building sessions with several sexual and gender minority groups. I found the article interesting to read although the language used sometimes is discouraging. I refer to the use of abbreviations that make the article unapproachable at times. Below are some suggestions for a systematic revision of the article.

I would encourage the authors to prepare a more informative introduction, in which the readers would be acquainted immediately not only of the broader issue but also of the method used (four groups, 27 individuals, etc), geographical localization of the study and the year of the study. The familiarization with the context is useful especially because healthcare systems are local. This is also needed in order to distinguish, if and when possible, between the regulatory framework that guides healthcare in relation to specific minority groups, and the subjective behaviour of healthcare workers that may be guided by their own status beliefs. I felt that the approach chosen by the authors does not allow us to see this distinction, but this should be somehow acknowledged in the introduction along with the scheme for the study.

In line with the previous comment, I would also like to ask the authors to introduce overall regulatory frameworks for minority groups in the specific region where the interviews were held. As I said, I miss the contextualization.

Another important element is to distinguish between disparities in health outcomes and in access because they represent two sides of a problem. Do the marginalized group feel excluded from accessing healthcare? Do they feel unequally treated instead in the (quality of) treatments in such way that their health outcomes are affected? I would ask the authors to divide clearly these two dimensions in their discussion for each dimension they treat.

Also, I encourage more nuancing in the type of healthcare (mental, primary, specialist) and some generalizations if possible for the type of care that is most affected by problems that sexual and minority groups face.

Overall, I missed somewhat more structure to the article and that refers to all its sections. The message that comes out now is important, but without the right contextualization and clarity, no clear policy implications can be made.

Minor

The abstract should also mention the type of health disparities studied (e.g. outcomes, access, type of care)

Attenuate the use of abbreviations and jargon

Comments on the Quality of English Language

My major complain is overly use of jargon. I have no other complaints

Reviewer 2 Report

Comments and Suggestions for Authors

Mapping the Dynamic Complexity of Sexual and Gender Minority Healthcare Disparities: A Systems Thinking Approach

1.     This research is focused on understanding and addressing health disparities experienced by Sexual and Gender Minority (SGM) populations compared to their heterosexual and cisgender counterparts. The paper emphasizes the importance of creating SGM-affirming and culturally responsive healthcare settings to address health disparities in SGM populations.

2.     Regarding participants, you mentioned that you have 27 participants. In Table 1, Age & Race (Ethnicity) have 27 participants, Sexuality has 26, and Gender-Identity has 28. 

3.     Please check Figure 2:  too anxious to go (line 348)

4.     In your Discussion, I suggest you refer the readers to your Figures. For example, in lines 424-429, you may refer the readers to Figures 4 and 5 to make the Discussion connected to the Results.

5.     The Figures in the Results are replete with significant data that are lost in the Discussion. You may consider exploring concepts you gave in the Results to be included in the Discussion.

6.     Research studies typically have limitations, and it is important to acknowledge them. Addressing the limitations of the research would provide a more balanced view of the study's results and conclusions.

7.     Any recommendations for future research or policy?

Comments on the Quality of English Language

There's a switch between past and present tense in the passage, which can be confusing. Maintaining consistent tense throughout the text is important for clarity.

Reviewer 3 Report

Comments and Suggestions for Authors

Mapping the Dynamic Complexity of Sexual and Gender Minority Healthcare Disparities: A Systems Thinking Approach

This is an interesting and important contribution regarding a relevant topic. Despite this, the article could benefit from several improvements before it can be accepted for publication, as it seems more a theoretical contribution than a research article:

1.     Authors spend a great amount of space in the introduction describing methodological aspects of CSS. I believe this should be put in the methodology section.

2.     The same applies to the intersectionality information provided. This should not be a theoretical base about intersectionality 3 paragraphs long.

3.     Also, more research stating other results from other sources is needed in the introduction.

4.     Objectives of this study must be clearly stated before the methodologies sections.

5.     I suggest authors to summarize information and / or send irrelevant information to a supplementary file.

6.     Discussion: author do not really discuss the results their results, comparing them to previous research.

7.     What is the real differentiation element to this approach?

8.     Limitations are not discussed.

9.     Implications are not discussed.

Best wishes.

Round 2

Reviewer 3 Report

Comments and Suggestions for Authors

Thank you for implementing all the requested changes to the manuscript. I believe the paper is now fit for publication. Best wishes.